# Perceptions of burnout among public sector physicians in Sierra Leone: A qualitative study

**Mohamed B. Jalloh**[1]*, **Asad Naveed**[2], **Sylnata A. A. Johnson**[3], **Abdul Karim Bah**[4], **Adesola G. Jegede**[5], **Fatmata B. Barrie**[5], **Amrit Virk**[6], **Arthur Sillah**[7]

**1** Department of Medicine, McMaster University, Hamilton, Ontario, Canada, **2** Division of General Surgery, St Michael's Hospital, Unity Health, Toronto, Ontario, Canada, **3** Sierra Leone Psychiatric Teaching Hospital, Freetown, Sierra Leone, **4** University of Sierra Leone Teaching Hospitals Complex, Freetown, Sierra Leone, **5** College of Medicine and Allied Health Sciences, University of Sierra Leone, Freetown, Sierra Leone, **6** School of Social and Political Science, University of Edinburgh, Edinburgh, United Kingdom, **7** School of Public Health, University of Washington, Seattle, Washington, United States of America

* jallom1@mcmaster.ca

**Data Availability Statement:** The full dataset is not made publicly available in a repository due to ethical restrictions. Ethical approval for the study was obtained from the Sierra Leone Ethics and

## Abstract

In Sierra Leone, physicians face a high risk of burnout due to systemic challenges, with studies suggesting a gap in recognizing and addressing this condition. We explored public-sector physicians' experiences and perceptions of the organizational structures and characteristics needed to help them thrive in a resource-limited practice setting. We conducted in-depth, semi-structured interviews with 24 public sector physicians across Western Area Urban (Freetown), Bo, Kono, and Kambia districts in Sierra Leone. Thematic content analysis was carried out using both deductive and inductive techniques to generate codes and identify key themes. Physicians in Sierra Leone face multifaceted challenges that significantly impact both healthcare delivery and personal well-being. Our findings reveal that overwhelming workload and stringent schedules contribute to burnout, directly compromising patient care quality. The emotional burden of caring for patients with economic constraints in accessing treatment further exacerbates physician stress. Limited resources, such as insufficient medical supplies and personnel, foster a sense of helplessness among clinicians, leading to detachment and cynicism towards their ability to effect change. In the absence of formal institutional support, physicians often rely on peer support to manage burnout. These challenges collectively undermine physicians' ability to provide optimal care, as the emotional and physical toll affects their decision-making and engagement with patients. Within Sierra Leone's resource-constrained healthcare context, systemic reforms are necessary to address the root causes of physician burnout, and to improve patient care. Our findings suggest that implementing formal support structures, including counselling services and mentorship programs, is crucial. Improving working conditions through better resource allocation and infrastructure development is essential. Developing strategies to address the emotional burden of care, including robust training programs, could enhance physician well-being, reduce burnout, and consequently improve the overall quality of patient care in Sierra Leone's public health sector.

Scientific Review Committee (SLESRC No.007/ 012024; email:efoday@mohs.gov.sl). When applying for ethical approval, the authors did not specify that the data would be publicly available in a repository. As part of the written and verbal consent, the authors assured participants that all data would be confidential and that access to the recordings would be restricted to the research team. The authors specified that "some of their words" may be used to report the findings of the study (included in the paper as non-identifiable quotes). However, making all raw data publicly available would be a breach of participants' ethical rights.

**Funding:** The authors received no specific funding for this work.

**Competing interests:** The authors have declared that no competing interests exist.

## Introduction

Burnout is a psychological condition characterized by emotional exhaustion, depersonalization, and a decreased sense of personal accomplishment and can be caused by chronic work-related stress [1]. While it is primarily studied in high-income countries [2–4], it may be more prevalent in low- and middle-income countries (LMICs), owing to limited resources and healthcare personnel. A study in Nigeria found that 46% of physicians suffer from burnout due to long working hours, insufficient pay, and unfavourable working conditions [5]. In Ghana, during the COVID-19 pandemic, 21% of the healthcare workers experienced burnout [6].

Burnout among healthcare professionals in Sierra Leone is a critical issue, particularly given the country's well-documented healthcare challenges [7]. A qualitative study conducted during the Ebola outbreak found that health care workers experienced high levels of stress, emotional exhaustion, and burnout [8]. Furthermore, another study examining public sector healthcare workers' motivation and retention in Sierra Leone revealed that low wages, poor working conditions, and limited opportunities for professional development contribute to diminished motivation and high attrition rates, which exacerbate burnout [9].

The psychological burden imposed on physicians by societal and systemic expectations significantly affects their ability to maintain professional commitment and personal well-being [10]. Symptoms of burnout are often mistaken for general work-related stress, highlighting a gap in the understanding and recognition of this issue [11]. This discrepancy suggests a lack of familiarity with burnout, underscoring the need for educational interventions to bridge the knowledge gap.

Organizational structure and characteristics are crucial for addressing physician burnout. Mitigating burnout should be integral to strategies aimed at improving health care systems [12]. Effective organizational strategies include locally developed modifications to clinical work processes [13]. Factors such as understaffing, lack of resources, difficult work schedules, inadequate job security, and poor salaries in public hospitals contribute significantly to burnout [14].

The impact of physician burnout extends beyond professional performance to personal life, leading to issues such as depression, suicidal ideation, substance abuse, and reduced work hours [15]. Moreover, burnout affects families, with spouses of burnout physicians experiencing secondary emotional trauma [16]. Addressing physician burnout is crucial not only for the well-being of healthcare professionals, but also for the effective functioning of healthcare systems.

In this study, we explored public sector physicians' experiences of burnout and their perceptions of the organizational structures and characteristics needed to help them thrive in a resource-limited practice setting. Understanding these perspectives is vital, not only for enhancing the well-being of healthcare professionals but also for informing policy actions and improving healthcare delivery systems.

## Methods

### Study setting

Sierra Leone, located on the west coast of Africa, is home to approximately 7.7 million people and had a Gross Domestic Product (GDP) per capita of US$1663 in 2018 (Purchasing Power Parity constant 2017 international $) [17]. More than half of the population lives below the poverty line, with an average life expectancy of 51 years at birth [17].

The country has a three-tier public healthcare system consisting of peripheral health units, 21 hospitals at the district level, and three specialist hospitals for referral cases. Furthermore,

there are 45 private clinics and 27 private hospitals, the majority of which are situated in the capital city of Freetown [17].

This study was conducted across four districts in Sierra Leone: Freetown (Western Area Urban), Bo (Southern province), Kono (Eastern province), and Kambia (Northwestern province). Freetown and Bo are two of the most densely populated out of the country's 16 districts. The country has fewer than 500 doctors including fewer than 100 specialists and consultants. A majority of these health professionals are concentrated in urban areas [9].

## Study design

This is an exploratory qualitative study employing in-depth structured interviews with 24 participants between April 1, 2022 and July 31, 2022. This study was designed through a series of discussions with research team members to ensure relevance to the larger project goals while retaining elements to support pragmatic analysis [18]. We adopted a pragmatic approach involving both deductive and inductive elements. However, we did not aim to produce a formal and generalizable theory. Instead, our study adopted a constructivist perspective, acknowledging that any analysis is shaped by the specific context of time, place, and situation [19]. We reported the study methods and results according to the Consolidated Criteria for Reporting Qualitative Research (COREQ) guidelines [20].

## Sampling

Participants were recruited purposively through maximum variation (mixture of men/women and career stages [house officers, medical officers, registrars, and various work regions]) to include those with first-hand knowledge and experience of service provision within public healthcare facilities in Sierra Leone. All the participants were over 18 years of age.

The interviews were conducted using predeveloped English interview guides. Two members of the study team, MBJ and either AKB or SAAJ – all of whom are medical doctors trained in qualitative interview methods – conducted each interview. One member led the interview while the other took notes. All interviews complied with the principle of informed consent, were audio-recorded, and lasted an average of 40 minutes.

The data sampling and collection process were carried out in collaboration with local physicians (AKB and SAAJ) and medical students (AGJ and FBB) who recruited participants and facilitated the interviews. We recruited a diverse sample of doctors from various departments, regions, and levels of experience (**Table 1**).

We ensured the trustworthiness of the data by adhering to health research standards of credibility, dependability, confirmability, and transferability [21]. To ensure credibility, long-term contacts with participants were maintained, extending up to a year when needed, to seek post-interview clarifications. Moreover, our research team was comprised of experienced researchers who collaboratively developed and fine-tuned the interview guide. Involving local researchers helped tailor data collection and analysis to the setting, ensuring context-sensitive questions and terminology. This enhanced the sense of study ownership among participants. The involvement of local researchers also helped balance power dynamics during the interviews, considering influencing social factors such as gender and work experience. With a local researcher present, participants were more at ease in discussing contentious or sensitive issues [22].

## Data analysis

Data analysis was informed by a general pragmatic approach, aligning emerging themes grounded in the data with pre-determined focal areas relevant to the overarching study

**Table 1. Characteristics of study participants.**

| Characteristic | Category | Total N (%) |
|---|---|---|
| Sex | Female | 12 (50.0) |
| | Male | 12 (50.0) |
| Age range (years) | 26–30 | 10 (41.7) |
| | 31–35 | 9 (37.5) |
| | 36–40 | 4 (16.7) |
| | >40 | 1 (4.2) |
| Designation | House officer (intern) | 5 (20.8) |
| | Medical officer | 11 (45.8) |
| | Registrar (resident) | 5 (20.8) |
| | Specialist | 2 (8.3) |
| | Consultant physician | 1 (4.2) |
| Location | Freetown | 16 (66.7) |
| | Kambia | 2 (8.3) |
| | Bo | 3 (12.5) |
| | Kono | 3 (12.5) |

objectives [18]. The audio recordings were transcribed verbatim into Microsoft Word and then imported into NVIVO software (version 12) to facilitate data analysis.

Two authors (MBJ and AKB) initially coded the transcripts independently, thereby forming a data framework featuring emerging themes and subthemes. Multiple strategies were used to enhance the validation of the data interpreted [19]. As we progressed with the coding, connections between categories emerged, facilitating a more theoretical coding approach. This method, paired with ongoing comparison, enabled a transition from descriptive to conceptual analysis. Consequently, we were able to construct a framework that guided further data sampling and coding, which enhanced our understanding of emerging themes and data saturation [23]. Efforts were made to check for 'deviant cases', which could potentially contradict the emerging themes. Finally, the data framework and draft narrative were shared with the co-authors, including those involved in data collection and the lead investigator to check for alignment with the study objectives, and collectively reviewed over several iterations.

## Participant and public involvement statement

The study design, conduct, and reporting did not involve participants or the public due to unforeseen delays and time constraints. However, we are considering a higher level of public and stakeholder engagement in disseminating the research findings.

## Ethical considerations

This study was approved and granted ethical clearance from the Sierra Leone Ethics and Scientific Review Committee (SLESRC No.007/012024). All participants gave their written informed consent before participating in the study. We ensured proper storage and management of the data. The recordings were deleted from the original devices and saved in password-protected files on both the computers and external hard drives. The transcripts and field notes were also deidentified and saved in a secure, password-protected file on both the computers and external hard drives.

## Results

**Table 1** summarizes the characteristics of all participants included in the analysis for this study. A total of 24 interviews were conducted across four districts in Sierra Leone, with an equal distribution of male and female participants. The majority of participants (41.7%) were between the ages of 26 and 30. Medical officers comprised the largest group of participants interviewed (45.8%), followed by house officers (interns) (20.8%), and registrars (residents) (20.8%). The majority of interviews were conducted in Freetown (66.7%) (**Table 1).**

Our study revealed several unique themes, sequentially discussed in the sections that follow:

### Workload and work schedule

Many clinicians reported that excess workload and tight work schedules had an impact on them. This was not merely an issue of personal wellness, but it extends to the quality of patient care and the overall efficacy of healthcare systems. One resident expressed:

"Well, it's huge, really, because we are charged with lot of responsibilities. Especially when we are on call. We have to clerk every patient that comes to the hospital in detail. Make sure we get treatment plans for them, review them throughout the day, make sure they are okay. And then when consultants come back, we have to review the patient meaning consultants make corrections. It's very hectic, when you're on call, it gets worse because you have to stay in the hospital for about two to three days, depending on how the call last. . .. You hardly sleep because at any time when you wake up in the middle of the night you see patients, it's very hectic."–(**Male, PB006**)

A house officer added:

"It's a busy job, yes, it is and yes there are times that I feel overburdened, there are times I feel like catching a break, feel like am hungry, but you can't get up to go and eat because you feel like you haven't done yet. At times I feel overburdened because it's not an easy job and there are not enough people to do it, yes that happen.–(**Female, PB002**)

This underscores the relentlessness of physicians' work life and the detrimental impact it can have on their wellbeing. The burnout issue is intensified by understaffing and increased workload, as articulated by a medical officer from a secondary facility in Freetown:

". . .there are four doctors in the facility, two work in Paediatrics, two in Obstetrics and Gynaecology and there is one who rotates in all departments between the outpatient and female medical ward. It's a hospital that focuses mainly on maternal and under five care but we have a female medical ward and an outpatient, so whenever you are not working in Paediatrics, you are rotating in these departments, so like basically you are working most of the time like the ward as to covered 24/7 so even if you are not at the hospital they will call you and say, so even if you are not working, you are working so it's feel as if you are working all the time."–(**Female, PB017**)

### Challenges in patient care and resource constraints

Physicians highlighted the difficulty of dealing with patients who cannot afford treatment or necessary tests, the limitations of the local healthcare system, and the emotional strain of witnessing patients' suffering. The severe poverty faced by many, combined with the obligation to

pay for out-of-pocket care, frequently results in patients' inability to afford the recommended treatment. Adding to this burden is the registration fee that all patients are required to pay upon hospital arrival. This financial issue contributes to compassion fatigue and burnout as clinicians strive to balance optimal care with patients' monetary constraints.

"It's either they're not able to afford the treatment because you're in a public sector and most of the people who come to the public sector hospitals are literally poor. . ."–**(Male, PB015)**

A medical officer in Freetown added:

". . .but the lack of resources can be very, very frustrating and like then that can tear you apart. In the fact that if you see a patient suffering, for example, you see you have a critical patient who is on oxygen. They're not like in the blink of an eye. The electricity just went out like that. And you see that patient struggling for breath. And you can't do much there's like not involved because sometimes even the backup generators that we have in the facilities, they will tell you that oh, we don't have like fuel we don't have reserved fuel to power these generators. . .. So sometimes you lose patients. Yeah. So, the only thing they will tell you oh yeah, you save some you lose some you understand, but these are all preventable factors. So, if you can't really, if we can't really like work in an environment where we have adequate resources or equipment, then it can be really overwhelming for the doctors. . ."–**(Male, PB013)**

Additionally, interviewees discussed issues like lack of supplies (gloves, sterile gauze), inefficient system (delays in obtaining blood for emergencies), and the resulting negative impact on patient care and physician well-being.

"I had to check the blood sugar. . .the glucose level of the patient and I needed a glucometer. I looked all round, there was no glucometer. . ."–**(Male, PB003)**

## Detachment and cynicism towards work

This theme came out strongly among many participants, often cited as a significant component of physician burnout. This concept is characterized by a pervasive negative attitude towards one's job, a loss of interest in work-related activities, and a reduced sense of personal accomplishment.

One resident encapsulates this sentiment:

"Yes, yes. I mean. I mean before I think when I was an intern, I think it was the first time I witnessed someone passing away, and I burst into tears. I was crying, and all people could say were like, I mean you just. . . eventually you get to a point where you don't even care about it."–**(Female, PB009)**

The inherent nature of the medical profession—the persistent exposure to stress, the demands for perfection and accountability, and the relentless working hours—can engender a sense of cynicism and detachment. This is more than just simple job dissatisfaction; it is a profound disconnection from one's work that can have significant ramifications on both personal well-being and patient care.

A medical officer working in Kambia district hospital reflects on this:

"It is exhausting. . . sometimes, you have frustrating days. Some days you get angry from the start of the day. You're angry with everything and everyone. Some days are so frustrating, you cannot get yourself that you're gonna wish that you're not even here. But the reality is you are here, like you are the only source of hope in terms of medical provision; you and your team. So, we have no option, we have no choice."–**(Male, PB007)**

Another medical officer added:

"It feels like I'm going through the motions, detached from the real purpose of my work. The frustration, the endless paperwork, the administrative burdens—it's hard to remember why I started this in the first place."–**(Female, PB020)**

This sense of detachment is not an individual failing, but rather an indication of systemic issues within the healthcare industry that need to be addressed.

"We have to sit back and watch our patients, well to say perish. . . and we cannot really help in anything."–**(Male, PB015)**

### Lack of institutional support and coping mechanisms

The respondents express the absence of formal support systems within the hospital for dealing with stress and burnout.

Many respondents explain how they use peer support as a coping mechanism, talking and deliberating on their shared difficulties.

"So, since all of us are frustrated together, we kind of . . . like serve as . . . like therapists for one another."–**(Male, PB004)**

"Nobody cares, nobody comes to their aid, and in fact when we have young people with mental illness, people will be ascribing it to drug abuse or other things instead of actually looking at other possibilities like work environment."–**(Male, PB012)**

"The work I do, I work as a doctor and I have a lot of stress in that work, psychological stress. . . that should not be personal, it has to be institutionalized."–**(Female, PB001)**

### Physicians' recommendations for change

The issue of physician burnout calls for strategic interventions, as identified by the physicians in our study. Key focus areas include the creation of a more favourable working environment that supports both the mental and physical well-being of physicians. Financial incentives and appropriate compensation for overtime work are essential for recognizing and motivating extra effort. Effective policy implementation and retention strategies are required to ensure a stable and supportive professional environment. Establishing structured work hours and robust institutional support is crucial for preventing overwork and managing workplace stress. Developing communication platforms offers physicians a space to share challenges and gain communal support, thereby reducing feelings of isolation. Enhancing the recognition of physicians' contributions and performance appreciation is fundamental to boosting morale and job satisfaction and addressing the critical issue of burnout in healthcare. It is crucial to implement strategies that guarantee the long-term viability of the recommendations outlined below.

**Improved working environment.**   Physicians emphasize the urgent need for improved work environments. As stated by one house officer:

"And the environment, we hope for a better environment. . . these are things I want to recommend, I want to appeal that the authorities look into, to improve on them."–**(Male, PB003)**

An environment conducive to mental and physical well-being can significantly reduce stress and burnout. This involves addressing factors such as workplace safety, adequate resources, and a supportive atmosphere.

**Financial incentives and overtime compensation.**   The lack of adequate compensation, particularly for overtime work, is a significant contributor to burnout. One resident suggests:

"If you have to work overtime, you should be motivated. You can have an incentive for that. I think that incentive can give you the extra energy to keep on pushing."–**(Female, PB005)**

Incentivizing overtime work can provide financial support and recognition for the additional efforts made by physicians, potentially enhancing morale and reducing burnout.

**Policy and retention plans.**   Physicians are calling for political attention to retention strategies, highlighting the need for well-structured policies to retain trained professionals:

"So, it's a thing that you really need political attention to look into because you train lots of people and then they will leave because there's no proper retention plan in place."–**(Male, PB015)**

Effective retention plans may include career development opportunities, supportive leadership, and policies that acknowledge and address the unique challenges faced by physicians.

**Structured work hours and institutional support.**   Overwork is a prevalent issue contributing to burnout. A suggestion to tackle this is:

"Let us start with the institutions, developing systems that will take care of their welfare plus have a structure of how many hours to work."–**(Male, PB012)**

Structured work hours can prevent chronic overwork, while institutional support systems can provide resources and assistance to manage workplace stress.

**Platforms for communication and support.**   Creating platforms for physicians to share experiences and challenges is crucial:

"So, let us have a platform that is good for all of us, and we can go and share our problems and issues with, and then discuss sometimes by just talking to some of them, it takes away about 50 to 70% of the burden of some of them."–**(Male, PB011)**

Such platforms can foster a sense of community, provide emotional support, and offer solutions to common problems, significantly reducing the sense of isolation that can accompany physician burnout.

**Recognition and performance appreciation.**   Finally, acknowledging and appreciating the work of physicians is vital:

"Salaries, performance approval, and then appreciating works. . . those are the kind of things that will help improve the system."–**(Female, PB021)**

Recognition, whether through financial rewards, career advancement opportunities, or simple acknowledgments, can boost morale and job satisfaction, which are essential in combating burnout.

## Discussion

This study explored the experiences and perceptions of burnout among public-sector physicians in Sierra Leone, given the paucity of research in settings with limited resources. Despite the increasing awareness of physician burnout, the complex interplay of poorly understood barriers presents a unique challenge (**Fig 1**). These barriers have significant implications for individual well-being and effectiveness of healthcare delivery.

Several themes emerged: the impact of heavy workloads and demanding schedules, obstacles encountered in patient care due to resource constraints, necessity of policy and retention strategies, growing sense of detachment and cynicism among physicians towards their work, and a lack of institutional support and coping strategies.

### Healthcare worker workload and patient care quality

Our study has identified several concerns regarding clinician workload and tight schedules, which aligns with recent literature. Accumulated evidence indicates that high workloads, particularly in inpatient settings, are associated with adverse outcomes, including increased hospital length of stay, delayed discharge, increased costs, and negatively impacted quality improvement efforts [24]. These factors contribute to clinician burnout, a global issue in the healthcare system.

The well-being of healthcare workers is crucial as it directly impacts the quality of patient care. Excessive work-related stress can lead to physical, psychological, and behavioural complications in physicians, which can compromise patient care [25]. The increased workload for hospital physicians due to residency work-hour restrictions and efforts to improve patient throughput further exacerbates this issue, demanding that hospitalists function in various capacities, potentially beyond their capabilities [26].

Furthermore, long working hours and extended shift durations for senior resident physicians have been associated with adverse patient and physician safety outcomes [27]. Preventable medical errors, which contribute to a significant number of patient deaths annually, have been linked to an excess clinical workload among resident physicians [26]. Insights from other healthcare professionals further support the detrimental effects of high workloads on patient care, highlighting the need for effective workload management strategies [28]. Healthcare workers under work overload are more likely to experience burnout and express intentions to leave their jobs, which can lead to understaffing and perpetuating the cycle of excess workload and decreased care quality [2]. These findings underscore the importance of addressing workload and work schedule issues not just from a wellness perspective but also as a critical component of maintaining patient care quality and efficacy within healthcare systems.

Our study revealed a critical issue that requires attention, namely, the absence of institutional support and coping mechanisms for physicians, which has been widely recognized in the literature. The increased psychological burden faced by healthcare workers due to demanding workloads and insufficient resources emphasizes the need for measures to build resilience and coping skills [10]. One study emphasized the necessity of theory-based interventions and supportive leadership to foster social support among healthcare workers [29]. This suggests

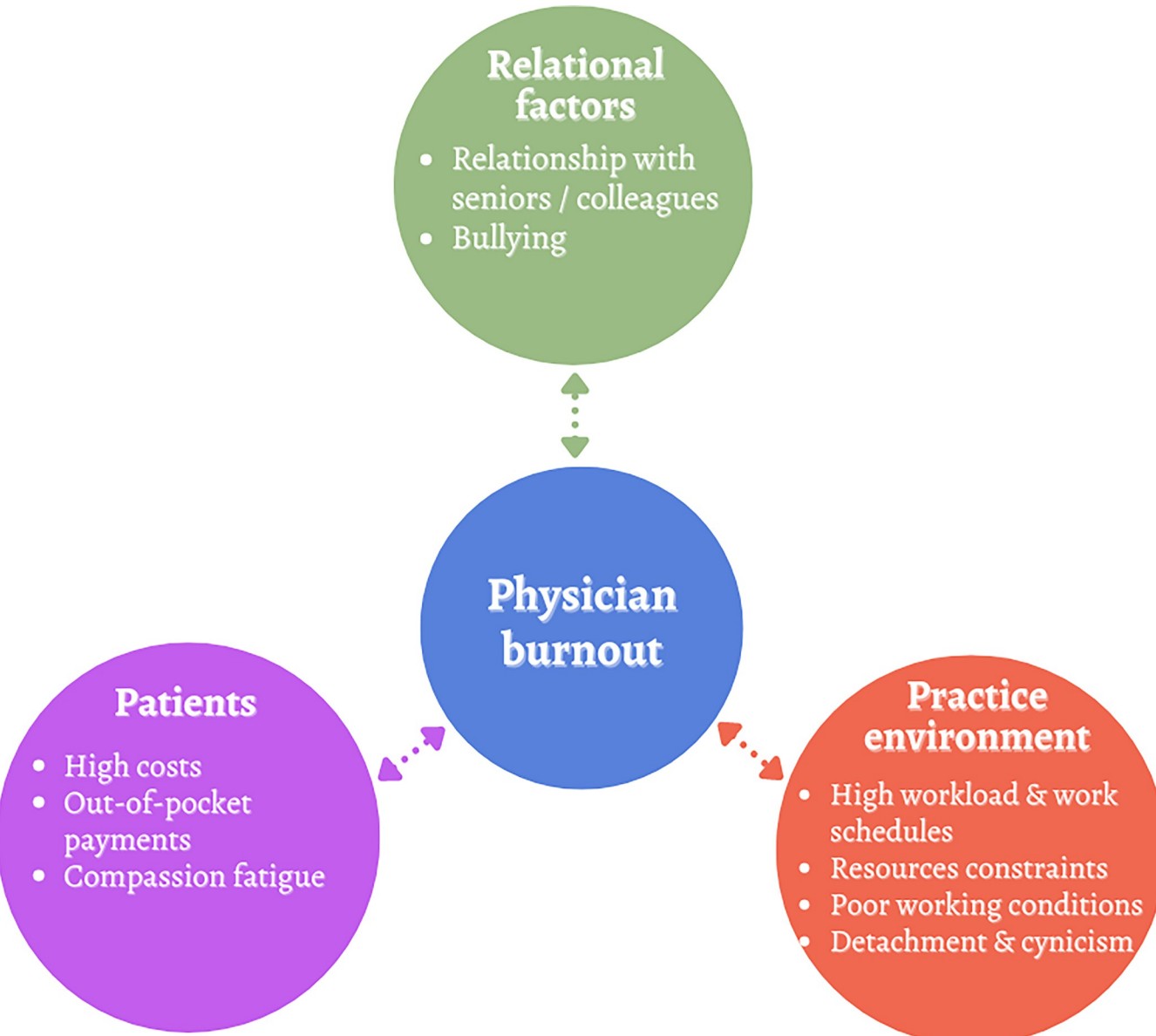

**Fig 1. Interconnected determinants of physician burnout.** The figure presents an integrative model identifying three core contributors to physician burnout —relational factors, patient financial burdens, and practice environment stressors—each depicted as interconnected elements influencing the central issue.

that institutional support systems should be structured and implemented to improve the mental health of healthcare workers [29]. Without proper support, the stressors experienced by healthcare workers can lead to a range of mental health issues, including depression, anxiety, and insomnia, particularly in regions such as sub-Saharan Africa during the COVID-19 pandemic [30].

Physicians in Sierra Leone use various strategies to combat burnout, such as prioritizing physical well-being and seeking clinical variety, which reflect a deep-seated desire for knowledge and skill advancement. This drive for professional development can be linked to career progression, highlighting the need for bespoke training programs tailored to the needs of different healthcare settings and specialties. Establishing such programs could provide the dual

benefit of enhancing career satisfaction and mitigating burnout [31]. Additionally, changes in institutional policies, particularly during public health emergencies, significantly affect the mental health of healthcare workers [6, 32]. Thus, practical recommendations aimed at fostering resilience and mental well-being are crucial, especially for adapting to evolving healthcare challenges [33].

Our study reflects the well-documented global trend of physician burnout characterized by detachment and cynicism. This state, emerging from prolonged stress and not indicative of personal failure, evolves into a form of organizational cynicism, a critical view of institutional motives and values [34, 35]. This pervasive condition transcends job dissatisfaction and extended working hours. Physicians, especially residents, experience decreased empathy, potentially leading to diminished care quality and heightened perception of medical errors [36]. Notably, in settings such as emergency departments, professionalism endures despite the growing cynicism and eroding empathy [37]. This underlines the need for systemic solutions to address the root causes of burnout, thereby safeguarding both health care providers' well-being and patient care standards.

## Organizational factors

This study indicates that physicians are acutely aware of the organizational roots that contribute to burnout. However, there is a prevalent tendency to prioritize personal protective strategies over systemic solutions. This inclination stems from the long-standing cultural image of physicians as being perfect and unwavering, which often leads them to downplay their own vulnerability, traits that are known to predispose them to burnout, and lead them to seek individualized solutions [38]. Physicians often feel uncertain about who should tackle organizational factors, and this sense of losing control over their professional environment not only fuels burnout, but also hinders their capacity to change at the organizational level [2].

Additionally, there is a noticeable lack of clarity among physicians regarding the effective confrontation of organizational challenges [39]. The erosion of control is identified as a key factor contributing to burnout among physicians [40]. This loss of control is often linked to external factors beyond their influence, leading to a sense of powerlessness [41]. The lack of autonomy due to various organizational challenges acts as a double-edged sword, exacerbating burnout and hindering effective interventions [15]. Physicians feel caught in a cycle where the erosion of control not only contributes to their burnout, but also diminishes their capacity to implement necessary changes within the organization [39].

Introducing monetary incentives for overtime within the social organization of medical work has broader implications. While financial incentives can provide immediate support and recognition [42], it is crucial to implement these measures cautiously to avoid inadvertently contributing to the institutionalization of burnout [43]. A balanced approach that includes both financial and non-financial support, such as effective workload management, creation of supportive work environments, and provision of professional development opportunities. Addressing these areas comprehensively is crucial to ensure the long-term well-being of physicians.

## Relational factors

Our study highlights the impact of relational factors on physician burnout in Sierra Leone. The results suggest that positive colleague interactions serve as a buffer against burnout, whereas isolation and unsupportive supervision exacerbate it. This aligns with earlier findings that emphasize the benefits of coaching and mentoring in talent development among medical trainees [44]. Indeed, participants highlighted the significance of peer support in alleviating

burnout symptoms, reinforcing the notion that supportive supervision models, including mentorship, can be effective interventions. Addressing excessive working hours is crucial for both patient safety and physician wellbeing [27]. Implementing a comprehensive approach that combines mentorship, supportive supervision, and manageable work hours can effectively reduce burnout by addressing both long working hours and deeper relational factors contributing to physician burnout.

## Strengths and limitations

Our study is the first to explore a deeper understanding of physicians' perception of burnout in Sierra Leone. Selecting a diverse group of doctors based on sex, age, and demographics allowed us to capture a wide range of perspectives. Although the sample size was small, with only 24 participants, Patton noted that qualitative research often focuses on small, specifically chosen samples, sometimes as few as one subject, to explore a particular issue in depth [45]. In our case, data saturation was achieved after approximately 20 interviews, with four additional interviews for assurance. However, we recognize that different contexts may reveal different experiences.

The present study is not without limitations. The recruitment of study participants through gatekeepers may have impacted their willingness to participate due to their relationships with these gatekeepers, potentially leading to researcher or participant bias. To mitigate this, in-country investigators only attended participant interviews when necessary. Data analysis and reporting were conducted jointly and iteratively by two authors (MBJ and AKB) until consensus was reached. It is important to note that the results of this qualitative study are based on the perceptions of the interviewed public sector physicians and may not be generalizable to all healthcare workers in Sierra Leone or other contexts. These findings provide valuable insights into the specific experiences of the participants, which can inform targeted interventions in similar settings.

## Conclusion

In this study, we aimed to explore the experiences and perceptions of burnout among public-sector physicians in Sierra Leone. We found that organizational and relational factors, including heavy workloads, resource constraints, and lack of institutional support, significantly contribute to burnout. Detailed accounts from interviewed physicians highlighted the relentless nature of their work schedules, the emotional and professional toll of inadequate resources, and the urgent need for systemic changes to improve their work environment and overall well-being. To mitigate the effects of burnout, policymakers should adopt a multifaceted approach. This involves investing in the healthcare workforce to alleviate workload, providing adequate necessary resources to reduce stress and boost morale, and empowering workers through increased autonomy. Furthermore, promoting supportive and respectful work culture is of utmost importance. In resource-constrained settings, systemic reforms are necessary to improve working conditions and develop strong support networks. These measures not only reduce the burden on healthcare providers, but also enhance their well-being and the quality of patient care they deliver.

## Supporting information

**S1 Checklist. Inclusivity in global research.**
(DOCX)

## Author Contributions

**Conceptualization:** Mohamed B. Jalloh, Arthur Sillah.

**Data curation:** Mohamed B. Jalloh, Sylnata A. A. Johnson, Abdul Karim Bah, Adesola G. Jegede, Fatmata B. Barrie.

**Formal analysis:** Mohamed B. Jalloh, Sylnata A. A. Johnson, Abdul Karim Bah, Adesola G. Jegede, Amrit Virk.

**Investigation:** Mohamed B. Jalloh, Sylnata A. A. Johnson.

**Methodology:** Mohamed B. Jalloh, Asad Naveed, Sylnata A. A. Johnson, Abdul Karim Bah, Arthur Sillah.

**Project administration:** Mohamed B. Jalloh.

**Software:** Mohamed B. Jalloh.

**Supervision:** Mohamed B. Jalloh, Arthur Sillah.

**Validation:** Mohamed B. Jalloh, Sylnata A. A. Johnson, Abdul Karim Bah, Adesola G. Jegede, Fatmata B. Barrie, Amrit Virk, Arthur Sillah.

**Visualization:** Mohamed B. Jalloh.

**Writing – original draft:** Mohamed B. Jalloh, Asad Naveed, Sylnata A. A. Johnson, Abdul Karim Bah, Adesola G. Jegede, Fatmata B. Barrie, Amrit Virk, Arthur Sillah.

**Writing – review & editing:** Mohamed B. Jalloh, Asad Naveed, Sylnata A. A. Johnson, Amrit Virk, Arthur Sillah.

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
