## [Decision Letter · Decision Letter 0]

14 May 2024

PGPH-D-24-00567

Health Workers’ Conceptualization and Perceptions About Burnout in  Sierra Leone: A Qualitative Analysis Of Public Sector Physicians

Dear Dr. Jalloh,

Thank you for submitting your manuscript to PLOS Global Public Health. After careful consideration, we feel that it has merit but does not fully meet PLOS Global Public Health’s publication criteria as it currently stands. Therefore, we invite you to submit a revised version of the manuscript that addresses the points raised during the review process.

We look forward to receiving your revised manuscript.

Kind regards,

Behdin Nowrouzi-Kia

Academic Editor

Journal Requirements:

2. Please include a complete copy of PLOS’ questionnaire on inclusivity in global research in your revised manuscript. Our policy for research in this area aims to improve transparency in the reporting of research performed outside of researchers’ own country or community. The policy applies to researchers who have travelled to a different country to conduct research, research with Indigenous populations or their lands, and research on cultural artefacts. The questionnaire can also be requested at the journal’s discretion for any other submissions, even if these conditions are not met.  Please find more information on the policy and a link to download a blank copy of the questionnaire here: https://journals.plos.org/globalpublichealth/s/best-practices-in-research-reporting. Please upload a completed version of your questionnaire as Supporting Information when you resubmit your manuscript.

3. Please provide separate figure files in .tif or .eps format only and remove any figures embedded in your manuscript file. Please also ensure all files are under our size limit of 10MB.

Additional Editor Comments (if provided):

The reviewer has suggested you provide minor revisions before the article is considered for publication

Reviewers' comments:

Reviewer's Responses to Questions

**Comments to the Author**

1. Does this manuscript meet PLOS Global Public Health’s publication criteria? Is the manuscript technically sound, and do the data support the conclusions? The manuscript must describe methodologically and ethically rigorous research with conclusions that are appropriately drawn based on the data presented.

Reviewer #1: Yes

Reviewer #2: Partly

2. Has the statistical analysis been performed appropriately and rigorously?

Reviewer #1: Yes

Reviewer #2: N/A

3. Have the authors made all data underlying the findings in their manuscript fully available (please refer to the Data Availability Statement at the start of the manuscript PDF file)?

Reviewer #1: Yes

Reviewer #2: Yes

4. Is the manuscript presented in an intelligible fashion and written in standard English?

Reviewer #1: Yes

Reviewer #2: Yes

5. Review Comments to the Author

Reviewer #1: I examined the article with great interest; it's intriguing. The authors have endeavoured to present a qualitative study of the problem of burnout that is of interest to the scientific community in the context of a country with limited resources.

There are aspects to which the authors can adhere in order to improve the quality of their article.

1- In the ‘abstract’, it is preferable to underline these points:

+ add keywords, such as perception and contextualisation, to make it easier to characterise and index the study in bibliographic databases.

2- With regard to the ‘introduction’ section, a number of points could be highlighted:

+ improve the general data section on the subject with more references, which is limited. And in any case, more recent literature. Ideally, from 2 to 5 years ago.

3. In the ‘discussion’ section, the following points can be put forward to improve the quality of the article:

+ It is preferable to enrich the discussion with more references on the various themes raised by the study.

Reviewer #2: About the title.

The title of the article announces two components - objectives: conceptualization and doctors' perceptions, however, the article does not present conceptualization information in the results, in the discussion or in the conclusion.

In the introduction of the article the concept that served as the basis for this qualitative study is cited and in Study Design of methods it is said: “We adopted a pragmatic approach involving both deductive and inductive elements. However, we did not aim to produce a formal and generalizable theory. Then, in Data Analysis: “Multiple strategies were used to enhance the validation of the data interpreted. (14) As we progressed with the coding, connections between categories emerged, facilitating a more theoretical coding approach. This method, paired with ongoing comparison, enabled a transition from descriptive to conceptual analysis. Consequently, we were able to construct a framework that guided further data sampling and coding, which enhanced our understanding of emerging themes and data saturation.” This description informs us how the researchers organized the sample, the interviews and the responses of the interviewees, for the analysis and presentation of the results, confirming that the qualitative study was focused on perceptions.

Due to the above, I suggest reviewing the title: “Conceptualization and perceptions of health workers on burnout in Sierra Leone: a qualitative analysis of public sector doctors”, I propose as an example: Qualitative study of Perceptions on burnout in the work of public sector doctors in Sierra Leone. 2022.

In the results I suggest checking:

Physicians Recommendations for Change, “Financial incentives and appropriate compensation for overtime work are essential for recognizing and motivating extra effort.” (...) It is crucial to implement strategies that guarantee the long-term viability of the recommendations outlined below.” And in: Financial Incentives and Overtime Compensation: which contains the researchers' analysis that appears in the article as: “Incentivizing overtime work can provide financial support and recognition for the additional efforts made by physicians, potentially enhancing morale and reducing burnout.”, because executing these statements would produce, that the monetary payment of overtime work the institutionalization of burnout, in contradiction of all the findings of the study.

In conclusion I propose to take into account in the writing:

The writing of the conclusion must compare the objectives and results based on the list of perceptions obtained from the interviewed doctors who describe explanations, demands and aspirations from their work experience.

The results of a qualitative study cannot be generalized as the perceptions of workers or doctors in the sector are restricted to the perceptions of the public sector doctors interviewed.

Perceptions describe a situation resulting from a set of labor relations in a health system with limitations and scope. More than a risk, it is a result.

The risk is to maintain the situation until it is no longer temporary and becomes a characteristic of the institution or system that serves the population in conditions of poverty.

The social organization of medical work should not include monetary incentives to work overtime because it would negate the organizational and relational factors that produce job burnout.

6. PLOS authors have the option to publish the peer review history of their article (what does this mean?). If published, this will include your full peer review and any attached files.

**Do you want your identity to be public for this peer review?** For information about this choice, including consent withdrawal, please see our Privacy Policy.

Reviewer #1: **Yes: **Chadrack KABEYA DIYOKA

Reviewer #2: No

---

## [Decision Letter · Decision Letter 1]

13 Aug 2024

PGPH-D-24-00567R1

Perceptions of Burnout Among Public Sector Physicians in Sierra Leone: A Qualitative Study

Dear Dr. Jalloh,

Thank you for submitting your manuscript to PLOS Global Public Health. After careful consideration, we feel that it has merit but does not fully meet PLOS Global Public Health’s publication criteria as it currently stands. Therefore, we invite you to submit a revised version of the manuscript that addresses the points raised during the review process.

Please address the minor outstanding suggestions made by Reviewer #2.

We look forward to receiving your revised manuscript.

Kind regards,

Avanti Dey, PhD

Staff Editor

Journal Requirements:

Additional Editor Comments (if provided):

Reviewers' comments:

Reviewer's Responses to Questions

**Comments to the Author**

1. If the authors have adequately addressed your comments raised in a previous round of review and you feel that this manuscript is now acceptable for publication, you may indicate that here to bypass the “Comments to the Author” section, enter your conflict of interest statement in the “Confidential to Editor” section, and submit your "Accept" recommendation.

Reviewer #1: All comments have been addressed

Reviewer #2: (No Response)

2. Does this manuscript meet PLOS Global Public Health’s publication criteria? Is the manuscript technically sound, and do the data support the conclusions? The manuscript must describe methodologically and ethically rigorous research with conclusions that are appropriately drawn based on the data presented.

Reviewer #1: Yes

Reviewer #2: Yes

3. Has the statistical analysis been performed appropriately and rigorously?

Reviewer #1: Yes

Reviewer #2: N/A

4. Have the authors made all data underlying the findings in their manuscript fully available (please refer to the Data Availability Statement at the start of the manuscript PDF file)?

Reviewer #1: Yes

Reviewer #2: Yes

5. Is the manuscript presented in an intelligible fashion and written in standard English?

Reviewer #1: Yes

Reviewer #2: Yes

6. Review Comments to the Author

Reviewer #1: Perceptions of Burnout Among Public Sector Physicians in Sierra Leone: A Qualitative Study

PGPH-D-24-00567R1

Following improvements to the manuscript quality, we are confident that it is technically sound and should be accepted for publication. This is because it provides valuable information that can contribute to the understanding of the organisational structures and characteristics necessary for public sector doctors to flourish in a country with limited resources.

Reviewer #2: I reviewed the abstract and the final version of the manuscript

Summary: Abstract

The title is changed correctly; however, the conclusions are recommendations of the researchers (of the authors of the article), it does not contrast the results of the research with the objectives of the research. It is striking to maintain in the conclusions “offer incentives for working overtime” when that is a minority expression among those interviewed.

Final manuscript

The title and text are maintained without the changes that are present in the summary.

7. PLOS authors have the option to publish the peer review history of their article (what does this mean?). If published, this will include your full peer review and any attached files.

**Do you want your identity to be public for this peer review?** For information about this choice, including consent withdrawal, please see our Privacy Policy.

Reviewer #1: **Yes: **Chadrack KABEYA DIYOKA

Reviewer #2: No

---

## [Editor Report · Decision Letter 2]

29 Aug 2024

Perceptions of Burnout Among Public Sector Physicians in Sierra Leone: A Qualitative Study

PGPH-D-24-00567R2

Dear Dr Jalloh,

We are pleased to inform you that your manuscript 'Perceptions of Burnout Among Public Sector Physicians in Sierra Leone: A Qualitative Study' has been provisionally accepted for publication in PLOS Global Public Health.

Best regards,

Julia Robinson

Executive Editor